# Feline Farmhands: The Value of Working Cats to Australian Dairy Farmers—A Case for Tax Deductibility

**DOI:** 10.3390/ani15060800

**Published:** 2025-03-12

**Authors:** Caitlin Crawford, Jacquie Rand, Olivia Forge, Vanessa Rohlf, Pauleen Bennett, Rebekah Scotney

**Affiliations:** 1Australian Pet Welfare Foundation, Kenmore, QLD 4069, Australia; j.rand@uq.edu.au; 2School of Veterinary Science, The University of Queensland, Gatton, QLD 4343, Australia; rebekah.scotney@uq.edu.au; 3The Far South Coast Branch, Animal Welfare League New South Wales, Bega, NSW 2550, Australia; oliviaforge@awlnsw.com.au; 4School of Psychology and Public Health, La Trobe University, Bendigo, VIC 3552, Australia; v.rohlf@latrobe.edu.au (V.R.); pauleen.bennett@latrobe.edu.au (P.B.)

**Keywords:** free-roaming cats, working animals, rodent control, Australian Tax Office, dairy farms, tax deductions, Australia, rodent overpopulation impacts, human wellbeing, animal welfare, feral cats, domestic cats

## Abstract

Rodents are viewed as pests on farms due to the damage they cause to machinery and produce and the health risks they pose to humans and livestock. Farmers often use poisons to control rodents; however, this method can be expensive, inefficient and pose a risk to wildlife, working dogs, children and pets. Domestic cats may offer a safer, cheaper and more efficient alternative for rodent control but there is concern about their impact on wildlife. This study sought to explore the views of Australian dairy farmers on the value of working cats on their farm, and their opinion on the Australian Tax Office making working cats’ care tax deductible. The results from 15 interviews (nine farms) revealed that working cats play an important role in pest control on dairy farms, saving farmers’ time and money whilst providing companionship. Farmers preferred having cats on the farm than using poison baits as they viewed the cats as safer, cheaper and more efficient and perceived them to have less impact on wildlife. Additionally, farmers strongly supported the care of working cats being tax deductible, stating that it would reduce financial pressure on farmers and improve cat welfare. It is recommended that the role of working cats on dairy farms be further explored. However, our findings suggest dairy farmers value having working cats on their farm and that the Australian Tax Office permitting their care to be tax deductible may benefit the wellbeing of dairy farmers and cats whilst protecting wildlife from exposure to poisons, toxoplasmosis and excess farm cats.

## 1. Introduction

### 1.1. Issues with Common Rodent Management Approaches

Agricultural settings often provide excellent habitats for rodents because of easy access to abundant food and suitable nesting areas. However, rodents cause immense damage to produce and crops because of consumption and through contamination with feces, urine and hair [1]. This can have severe economic implications for farmers [2]. Many infectious diseases spread to humans are zoonotic in origin [3], and rodents play a substantial role in the spread of these diseases to humans and livestock as they are vectors for leptospirosis, scrub typhus, gastrointestinal infections and rat-bite fever [4,5,6]. Moreover, invasive rodents have considerable negative impacts on natural ecosystems, particularly affecting ground-nesting birds on islands in the Pacific Ocean [7,8,9]. On today’s large and increasingly automated dairy farms, where technology use is highly advanced and complex [10,11,12,13], rodents’ gnawing habits can cause damage to wiring and other expensive infrastructure that is critical to farm operations [14,15]. Rats and mice are also prolific breeders, producing up to six litters per year, each averaging three to seven young [16,17].

For these reasons, it is clearly necessary to control rodent populations, particularly around farms where food is produced. Several control methods are available to farmers and can be claimed as a tax-deductible expense, including trapping, habitat management, repellents and rodenticides. Farmers often use rodenticides to control rat and mice numbers, as this requires little expertise and provides tangible results [7]. Rodenticides are lethal chemical agents, commonly referred to as rat poisons [18]. Most rodenticides are anticoagulants and act by preventing the process of blood clotting. They are categorized as either first-generation anticoagulants, where rodents must ingest multiple baits to consume a lethal dose, or second-generation anticoagulants, where rodents can consume a lethal dose in a single feed [19,20].

Whilst the use of rodenticides can be somewhat effective at reducing the rodent population around farm buildings, it can have a negative impact on native wildlife through primary and secondary poisoning [20,21,22,23]. Additionally, due to excessive use, rodent populations are becoming progressively more resistant to rodenticides [24,25,26], making their use increasingly less effective. Furthermore, whilst potentially being effective at decreasing the initial rodent population, using rodenticide baits requires ongoing work to keep rodents at bay, due to their high fecundity and short life cycle. This can be costly and time consuming for busy farmers [5,7,15], creating an additional hazard to an occupational group already at risk of negative psychological impacts due to financial worries and time constraints [27,28,29,30]. To overcome the issues with rodenticides, many farmers, including dairy farmers, use cats for rodent control [31,32].

### 1.2. Cats as Working Animals

Up until the mid-20th century, animals were commonly used to power farms [33]. In many countries, these animals have now been replaced with machinery; however, working animals still exist all over the world and are used for agricultural needs, as well as in the police force and as assistance animals for people with medical conditions [33,34,35,36]. These animals are seen as essential in countries like Australia, the United Kingdom (UK) and the United States of America (USA), being recognized as such by governments, who allow working animals’ care to be tax deductible as a business expense [37,38,39].

Cats were humans’ main protection from rodents, the diseases they carry and the damage they cause to produce and infrastructure for thousands of years [40] and remain a widely utilized method of pest control. A Scottish study found that cats were the most common form of non-chemical rodent control on farms growing crops [31]. Among the farms that used non-chemical rodent control methods, 65% used cats as an alternative, while 47% of those that did use rodenticides, had cats as an additional rodent control method [31]. The USA and UK still recognize cats’ roles in rodent control and permit them to be classified as working animals, making their care tax deductible [38,39,41]. However, in Australia, the government limits the classification of working animals to just dogs and horses [37]. Additionally, in most states of Australia, dogs and cats, including working dogs, must be microchipped and registered by 4 months of age, but the registration costs for working farm dogs are waived [42]. In contrast, in NSW, for example, initial registration for a cat is A$68 and if it is not sterilized by 4 months of age, another A$96 per year is payable, with no waiving of costs for working cats involved in rodent control on farms [42]. In addition, permits for keeping more than 2–4 cats are required in most states and range from A$50 to more than A$500, and in some councils, annual renewal fees exceed A$300, particularly for households with more than five cats [43,44,45]. This likely reflects concerns about cats’ impact on wildlife and the effects of various nuisance behaviors, such as fighting, defecating and producing unwanted litters [46,47,48], but it may be an important oversight. Due to these costs, it is unlikely most cats on farms are microchipped and registered with the local government, as even in urban areas, registration and permit costs are a barrier for low-income cat owners [49]. Permitting cats who are being used as pest control on farms to be tax deductible and waiving registration and permit fees may provide an incentive for farmers to microchip and register cats, as well as sterilize, vaccinate and provide them with food. Providing care for cats, such as sterilization and food, reduces their need to predate to meet their energy needs and reduces the population size over time, and hence reduces impacts on native wildlife [50,51]. Allowing care of working cats to be tax deductible could, therefore, provide farmers with an alternative to toxic baiting for rodent management, without putting further financial strain on them. Additionally, it would benefit the environment by making population management of cats more financially feasible for farmers and decreasing the use of rodenticides.

### 1.3. Gaps and Aims

A previous study showed that dairy farmers who had cats on their farm view them as valuable working animals rather than pets, indicating that cats were their preferred method of pest management [15]. Our study aimed to further explore the value dairy farmers place on cats on the farm and to document the opinions and potential impacts for farmers of the Australian Tax Office (ATO) classifying cats on farms as working cats. This would allow the cats’ care, including sterilization costs, to be tax deductible and local government registration and permit costs to be waived. Due to the amount of data collected in this study, this current paper reports the value of cats to dairy farmers and their views and arguments to the ATO regarding cats being classified as working animals. A second paper will report on farmers’ experience before and after participating in a free working cat sterilization program and their views on a Barn Cat Program. We anticipate that our findings will provide a foundation for other studies to build upon, helping to inform local government and welfare agencies of the role working cats play on farms. This, in turn, may potentially provide information that can allow cats on Australian farms to be classified as working cats, making their care tax deductible, like current regulations in countries such as the UK and US.

## 2. Materials and Methods

### 2.1. Research Design

To investigate the perspectives of dairy farmers regarding the cats on their farms and their experiences before and after participating in a free working cat sterilization program, a phenomenological approach was used. This approach allows the researchers to understand a phenomenon from the perspective of the people involved [52], therefore making this is an appropriate methodological approach. To be eligible for partaking in this study, participants had to be living and/or working on a dairy farm, be over the age of 18, have multiple cats on their farm, be residents of either city of Ipswich, Queensland (QLD), or Bega Valley, New South Wales (NSW), and have been through one of two cat sterilization programs (described below). Farmers were recruited to participate after they had cats sterilized through one of two free sterilization programs (described below). The first author (C.C.) conducted semi-structured interviews which were used to assist in gaining a greater understanding of the value of cats to dairy farmers and the impacts of a free working cat sterilization program on dairy farmers.

### 2.2. Participants

The 15 participants were all dairy farmers over the age of 18 years who had multiple cats on their farm and were residents of either Ipswich City, QLD, or Bega Valley, NSW. When the interviews took place, participants estimated that there were between 3 and 60 cats on their property, with an estimated mean, per farm, of 18 cats. This was a reduction from a mean of 44 cats (range 3–175 cats) immediately prior to the sterilization program as a result of the euthanasia of sick and excess cats and the adoption of kittens. For farms with multiple participants, the mean cat population estimate from each farm was used to calculate the overall mean number of cats per farm. Fourteen of the participants described themselves as being the owner of the cats (3–60 cats) and one participant stated that they did not own the cats (15 cats).

### 2.3. Procedure

The University of Queensland Human Ethics Committee (2024/HE000242) granted ethical approval before the study began. This study used purposive sampling [53] to contact farmers who had multiple cats on their farm and who had participated in a cat sterilization program. Participants agreed to be interviewed after they had been involved with one of two cat sterilization programs. Participation in the study was not a prerequisite to receiving free sterilization. One dairy farm was in the city of Ipswich, QLD, Australia, and the cats were sterilized as part of the Australian Community Cat Program, a research project initiated by the Australian Pet Welfare Foundation, Brisbane, Australia in collaboration with the Royal Society for Prevention of Cruelty to Animals, Queensland (RSPCA QLD), Brisbane, Australia and the Animal Welfare League Queensland (AWLQ) Gold Coast, Australia with funding and in-kind support from multiple organizations, including the Fondation Brigette Bardot (Paris, France) for sterilizations, MSD Animal Health (Macquarie Park, NSW, Australia) for vaccinations and parasite control, Central Animal Records (Keysborough, Australia) for the microchip registration of cats and Neighborhood Cats (New York, NY, USA) for cat traps and expert advice. This program provided free cat sterilization, microchipping, veterinary care for issues affecting their welfare, and endo- and ectoparasite control for cats being cared for by cat caregivers (semi-owners) or owners. Cats recruited to this program were permitted to be sterilized, microchipped, ear-tipped and returned to their original location without being legally registered to an owner and were instead categorized as Restricted Matter (approved by the Queensland Government under a Department of Agriculture and Fisheries Scientific Research Permit No. PRID000825). In place of the owner’s name or contact details, <suburb> Community Cat, and the Australian Pet Welfare Foundation name and mobile phone numbers were registered as secondary contact details on the microchip database of cats that were classified as restricted matter, including the personal mobile (cell) phone number of one author (J.R.) who was chief investigator on the project and a registered veterinarian. This phone number was used with the intention of providing a safety net for cats whereby there was someone who could be contacted to quickly decide on management of the cat if there was no owner to do so, especially if injured or sick. Microchip number, contact details, and address for the carer were recorded on an internal database accessible 24/7 by relevant APWF, RSPCA Qld, and AWLQ staff [54].

The second program, ‘The Far South Coast Dairy Cat Project’, took place in Bega Valley, NSW, Australia. In this program, dairy farmers in Bega Valley, NSW, were provided with free sterilization, preventative medication and vaccination of cats and were assisted in re-homing unwanted kittens. This program was undertaken by the Animal Welfare League New South Wales with further funding and support from the South East Local Land Services, the Far South Coast Landcare Association, local Bega veterinary practices, and a grant from the Office of Local Government, NSW. Cats were microchipped, with the farmer being listed as the owner on the state microchip database, but because of cost, were not registered, nor were other permit costs paid associated with cats being older than 4 months and being in excess of the numbers allowed to be owned without a permit.

Farmers who were eligible to participate were contacted by phone and email. Those who showed an interest in participating in the study were emailed a participant information sheet and consent form or were provided with a hard copy. Prospective participants were informed that participation was voluntary and confidential, and that should there be any question they were not comfortable answering, they did not have to answer and could withdraw from the study at any time.

Participants (15) from 9 of the 10 (90%) dairy farms involved in the cat sterilization programs agreed to be involved in the study, with 13 interviews taking place between the 5 April 2024 and the 8 May 2024. In two of the interviews, there were two participants, and the remaining interviews only had one participant. Semi-structured interviews were used to collect data. Nine interviews were conducted in person, with the remaining four being conducted over the phone. An Olympus Voice Recorder VN-541PC was used to voice record all interviews.

All interviews lasted between 29 min and 62 min (average 43 min); the duration of each entire interview included responses to questions that were not analyzed in this study but are being reported in a subsequent manuscript [55]. After completion of interviews, 11 interviews were transcribed by the first author (C.C.). The remaining two interviews, which had two interviewees each, were transcribed by a professional transcription service (Pacific Transcription Pty Ltd., Brisbane, Australia). The interviews form the basis of a larger study. However, this paper explores participant responses to a subset of questions focusing on what the cats’ purpose was on the farm and if there were any benefits to their presence; what the farmer’s experiences were without having cats on the farm (if relevant); and what their view was on cats being classified as working cats by the ATO and, if positive, how they would argue or justify this.

Thematic analysis was used to analyze the transcripts using an inductive approach search for recurring words or units of meaning, organize them into groups and themes and give voice to participant experiences [56,57]. Whilst certain themes arose due to the subject of certain questions, due to interviews being semi-structured, the text from the entire transcript was analyzed rather than each individual question, as follow-up questions often differed between interviews. Thematic analysis was performed using a six-phase approach [57]. Firstly, the author performing the initial analysis (C.C.) became familiar with the data by transcribing interviews, reading through the transcripts, and taking notes. The transcripts were then coded, with each unit of meaning being no longer than a sentence and no double coding being performed. A similar coding framework was used for all transcripts; however, due to the nature of semi-structured interviews, codes varied slightly between transcripts. These codes were then organized into corresponding themes and were reviewed and discussed with the research team. The interpretation of the data was then discussed amongst the research team until a consensus was reached and themes were defined and named. The outcome of the thematic analysis reported in this publication revealed nine main themes. Three related to pest control issues farmers had prior to acquiring cats (damage to infrastructure, human and animal health issues, dislike of rodenticides). Four themes were associated with the value of cats to farmers (pest control, monetary savings, companionship, monetary concerns for care of cats). The final two themes were related to farmers’ views on classifying cats as working animals (farmers’ views and farmers’ arguments to the ATO). All human names were omitted, and cat names were changed during the write-up of the results to maintain the confidentiality and privacy of participants.

## 3. Results and Discussion

Our results indicate that dairy farmers value having cats on their farm for several reasons, particularly for rodent control and the associated monetary savings, as well as for companionship. However, farmers stated that even though the cats were valued and having them on the farm was saving money, they felt care such as sterilization and vaccinations were not affordable. All farmers we spoke to felt positively about the ATO permitting cats to be tax deductible and many made strong arguments for doing so. These themes and sub-themes are discussed in the following section.

### 3.1. Pest Control Issues Prior to Cats

Ten of the fifteen (67%) dairy farmers interviewed in our study had experienced periods of time without cats on the farm and highlighted issues with pests such as rats, mice and snakes. Some of the five dairy farmers who reported always having cats on the farm discussed similar issues when the working cat population substantially decreased after culling. All dairy farmers interviewed discussed their concerns with having rodent pests around the dairy, including human health and food safety, damage to infrastructure and the cost of repairs. Participants who had been without cats for any period of time also discussed how they were previously managing pest populations, with many indicating that they were using rodenticide baits. However, most participants indicated that they had a negative view of rodenticide baits and would prefer not to use them. The sub-themes under this theme are discussed in the following section and tabulated with context examples in Table 1.

#### 3.1.1. Damage to Infrastructure and Feedstores

The damage that rats and mice caused to infrastructure was raised most frequently by farmers who had been previously without cats. Breakdowns in milking systems through chewed wiring was discussed as a major issue alongside damage to other machinery and the destruction of crops and feed for livestock.


*‘Well, we didn’t have a cat, and the rats chewed all the wire in the old dairy—just stripped the lot.’*



*‘[When] we got down to the low numbers [of cats], they [rodents] just got into the wiring and stuff to run the dairy, and you can’t have breakdowns in the dairy—it’s gotta go twice a day, every day.’*



*‘[Without the cats] obviously more equipment damaged and within a dairy like there’s a lot of wiring. So, the whole system is run by wiring, so it’s a nightmare when they [rodents] get into that and start chewing on the wires. Yeah, so it would be very frustrating.’*



*‘The rats and the mice get in and eat the feed that we had stored for the cows and seed that we had stored … like the feed for the cows you’d find mice and rats in the bags that are stored and they, yeah, they chew holes in bags.’*


Additionally, the damage to infrastructure often led to concerns regarding the cost of repairs. Participants explained that due to the nature of the machinery, the damage was often not something they could repair themselves. This meant that a technician or electrician would have to be called out to repair broken wiring, which participants said was costly. One participant mentioned that some machinery had to be sent away to be repaired by a specialist, resulting in that piece of machinery being out of use for several weeks. It was also expensive to repair.


*‘The cost of it and the money […] and the expense to fix everything—it’d drive you mad. It’d probably put you out of business if they [the rats] chew enough.’*



*‘Well, before the cats, we had a telly-handler eaten [by rats]. All the rubber hosing and the wiring, and it cost us about $45,000 to get that one machine going again.’*



*‘It was a costly exercise, so what would happen, because the dairy is all electronic, like with auto cup removers and the pulsation, it’s all electronic so all very fine wires too, so it was something that, because there’s a cluster of fine wires, it wasn’t easy for us as farmers to fix it, and we were forever calling that technician.’*


The damage that rodents can cause to crops and infrastructure is well documented [2,4,7,58]. Machinery breakdowns have been shown to be a stress factor for farmers [59,60], along with associated factors such as financial and time pressures. Machinery downtime caused by rodents damaging infrastructure and the cost of having machinery professionally repaired are likely to increase stress levels in dairy farmers, potentially leading to negative mental health impacts. Farmers have been documented to already be more at risk of poor mental health and have higher incidences of suicide than the general population [27,61,62,63]. Additionally, farmers highlighted that dairy cattle must be milked twice a day, every day for their welfare, which is also documented in the literature [64]. This suggests that the damage rodents cause to machinery not only negatively impacts farmer wellbeing, but also livestock welfare. There is little research conducted on rodents’ impacts on dairy farms specifically, but the comments made by dairy farmers in our study indicate that, due to the complicated nature of milking machinery, coupled with the welfare risk to dairy cattle during machinery breakdowns, rodent overpopulation may be more detrimental to the running of dairy farms and the farmers themselves than on other types of farms. However, this needs to be explored in other types of farming operations, for example, other operations which have expensive equipment with electrical wiring and rubber hosing, along with food stores that attract rodents.

#### 3.1.2. Human and Animal Health Issues

Participants explained that having rats and mice around the dairy created concern about food safety, with some discussing issues regarding the contamination of grain with rodent feces and the spread of disease to livestock.


*‘Mice, it’s probably just the contamination of the grain which feeds the cows. […] Contamination [via] like feces that sort of thing. Cows eat the feces and then cows get sick and just that health issue, plus the stink as well.’*



*‘The rats and the mice [get] in our feed for the cows. [We] didn’t want them getting in our feed for the cows, you know, spreading their diseases around.’*


Rodents are known to be vectors of disease to livestock and humans through feed contamination [65,66]. Additionally, rodents are a risk to food safety because they are carriers of a number of foodborne pathogens, such as *Salmonella* and *E*. *coli*. These can be passed through rodent droppings, contaminating food products and animal feed and potentially being passed through the whole food chain from animal feeds in primary production through to households and restaurants [67,68,69]. Controlling the rodent population, alongside good manure management, has been documented to reduce the transmission of pathogens, reducing the risk to livestock and food safety [66,70]. Because of this, dairy farms must control rodents to limit food safety hazards [71].

Farmers stated that without having cats on the farm they noticed more snakes around the farm buildings and expressed concern about the safety of their staff, family and pets.


*‘Every couple of days I would roll the plastic back to bring the new silage out, and underneath that plastic, there would be always dozens of rats and mice underneath that plastic. They would live on top of the silage, but under the plastic, and that was an absolute breeding ground for snakes because they just loved it, the food was on tap.’*



*‘Our welfare and our staffs’ welfare having snakes hanging around it’s not something that would be ideal.’*


Whilst snakes themselves can be viewed as a form of rodent control [72], they often pose a risk to humans, livestock, and domestic animals. Although very few people die from snake bites each year in Australia, hundreds are hospitalized [72,73]. Additionally, snake bites to domestic animals are frequent and often lethal [74]. Many of the farmers we interviewed discussed having pets and children around the farm, suggesting the risk that snakes pose to them is likely to outweigh any pest control benefits.

#### 3.1.3. Dislike of Rodenticides

When participants were asked how they were managing the rodent population prior to having cats, they explained that they used rodenticide baits, with one participant stating that they occasionally shot the rats. However, farmers indicated that they had a negative view of using rodenticides because of risks to wildlife, working dogs, pets, and children. Additionally, participants indicated that baiting was costly and ineffective, with some participants highlighting the potential risk to livestock and food safety from rats and mice dying and decomposing in unknown places around the dairy, including in animal feed.


*‘Baiting’s not great for the other wildlife, and we’ve got dogs and I’d prefer not to use the baits.’*



*‘Look, I’ve tried rat poisons and stuff, like that and it can get really expensive.’*



*‘When you poison them [rodents], they’d go and die somewhere. You couldn’t get them and then you smell a stink. Then you had maggots crawling everywhere, which is not good in a dairy.’*


Farmers’ comments and rationale for the risk rodenticides pose to wildlife, working dogs, pets, and children are supported in the literature. The negative impacts that rodenticides have on wildlife through primary and secondary poisoning have been well documented [19,20,75,76,77]. Because of the similar physiology of rodents and non-target mammals and birds, rodenticides can be just as hazardous to non-target species as they are to rodents [75,76]. Rodenticides pose a risk to children, with pesticides being one of the main causes of accidental poisonings in children, and incidences of poisonings are higher in rural areas [78]. Alternative methods of rodent control, such as biological control, have been mentioned as a potential countermeasure for rodenticide poisonings [79]. When looking into the use of biological control, recent focus has been on the use of avian predators [80,81]. However, there has been little research on how increasing the predatory bird populations impacts local wildlife, though one study indicated the introduction of barn owls had a negative impact on native seabirds in Hawaii [82]. Additionally, the contamination of stored feed by bird feces is a potential risk to livestock and therefore food safety [65]. Introducing another form of biological control without understanding the impact on native wildlife would not be advised.

The literature also supports the farmers’ comments regarding the cost and efficiency of baits [83,84]. Whilst some studies indicate the use of rodenticides can reduce the negative impacts of rodent populations on farms [85], the overuse of anticoagulant rodenticides around the world is leading to increased resistance in rodents [24,25,26,86]. This not only increases the strength of rodenticides needed to control rodent populations, and therefore the expense to farmers, but increases the risk to wildlife populations through secondary poisoning as rodents have higher levels of poison in their system. Additionally, most rodent pest species exhibit neophobia towards both novel objects and tastes, leading to low bait acceptance [84,87]. Moreover, if rodents die in inaccessible places or in livestock feed, it leads to an unpleasant work environment for farmers and poses a further risk to livestock and food safety. The use of lethal traps has been suggested as an effective alternative to rodenticide baits [85,88,89]. However, this method is labor-intensive and impractical when rodent populations are large, so farmers tend to be less likely to use this method [83,90]. Moreover, trapping must be continuous, with one study showing that rodents quickly returned to the site after the cessation of trapping [89].

With the issues surrounding rodent overpopulation and rodenticide use, there is a clear need for an alternative, effective method to control rodent populations in agricultural settings.

### 3.2. Value of Cats to Farmers

Participants stated that the cats’ purpose was pest control, with many indicating that the cats were a necessity on the farm. Most participants highlighted their preference for cats as pest management over other management options, such as rodenticide baiting. When discussing the benefits of having cats on the farm, three main sub-themes arose: a reduction in pests such as rodents and snakes, monetary savings, and companionship. The themes and sub-themes are discussed in the following sections and tabulated with context examples in Table 2.

#### 3.2.1. Pest Control

Participants were asked if they could classify the cats’ purpose on the farm, and all of them indicated that the cats’ purpose on the farm was pest and vermin control, discussing that whilst they did provide some companionship, the cats had a purpose on the farm and ’a job to do’. When the cats were discussed further, many stated that cats were a necessity and that they would not be without them.


*‘Well, this is the thing, we do need cats at our dairy because all the electronics and the wiring. If rats get hold of that, they can do a lot of destruction. So, we do need cats at our dairy.’*



*‘We couldn’t do without them [the cats]. We wouldn’t do without them, now. Otherwise, you’d be overrun with rats.’*



*‘It’s a must to have them [the cats] when you know the job that they do; it’s a must to have them especially on our dairy farm.’*


Additionally, when discussing the cats’ purpose on the farm, many participants indicated that they preferred using cats as pest control over other forms, such as baiting, due to cats being safer and more efficient in reducing rodent numbers than rodenticide baits. Many used words such as “nicer”, “reliable” and “efficient” to describe the cats. Many participants also indicated that they had reduced or completely stopped the use of rodenticide baits since acquiring cats.


*‘Yeah, you can try rat poison and stuff like that, but it doesn’t seem as effective as the cats.’*



*‘The cats seem to work everyday where baits are only ever any good while you’ve got bait out. Like if they [rodents] come and eat it all, you’re not on top of that checking, like they can eat all your baits overnight and then you’ve got none. So, you’re only poisoned for that one night where the cats, […] they just work everyday of the year. […] I don’t know, it’s just reliable.’*



*‘Well, we don’t like the baits around cause of our dogs and little children. […] I’ve got a young, young one [child] with us, and he’s with us at the dairy all the time, and we would rather just have the cats than the bait.’*



*‘You still never got them [the rodents] though. Like you would still see rats and mice even when you had baits. So, there’s nothing, absolutely nothing as efficient as a dozen cats.’*


Participants indicated that having cats on the farm reduced the number of rodents around the dairy. Many participants suggested that they rarely saw any rats or mice, with one participant describing it as a novelty if they were to see a rat or a mouse around the diary.


*‘I guess between here, the dairy and the calf shed, we haven’t seen any mice, rats, nothing. So, yeah, that benefit is worth more than what the cats will ever be worth.’*



*‘Seven years since I think we started with the cats and we’ve never seen a rat or a mouse since. So they’re doing their job.’*



*‘There was rats and mice here for 50 years, basically all that I can remember. So yeah, so we always had rats and mice, but if we went searching on our farm [now] to try and find a mouse, I don’t believe we’d find one.’*


All farmers stated during their interviews that having cats on the farm was beneficial, and in their view, the most efficient way to control rodent populations. However, published evidence on the impact of cat populations on rodent populations is conflicting. Some studies indicate that cats are not an effective form of rodent control, with their findings showing that cats only target small or juvenile rats [91,92], with one study finding no relationship between cat and rat populations [92]. Additionally, another study indicated that only the combined presence of cats and dogs deterred rodents from rural homesteads [93]. However, several studies indicate that cats are an effective form of rodent control, stating that the mere presence of cats or their scent deters rodents and suppresses reproduction [94,95,96]. Another study found that four farms introduced cats following initial rat extermination and remained rat-free, whereas one farm without cats experienced recurrent rat infestations [97]. Rodent control options such as trapping or the use of rodenticide baits were investigated in some studies [85,98] and were found to reduce the initial rodent population quickly. However, one of the issues with lethal rodent management is that, due to the high reproductive potential and compensatory response to severe population reduction, rodent populations can quickly recover if not continuously culled, meaning that the lethal management of rodent populations alone is rarely effective unless continuous, which may be impractical or too expensive for farmers [89,99,100]. Therefore, we hypothesize that the effectiveness of cats as a form of pest control, as described by the farmers in our study, may be attributed to the combination of the cats’ predation of rodents and their constant presence serving as a deterrent. Previous studies have generally focused on the impacts of cats on rodent populations in urban areas or laboratory environments [91,92,94,95,96]; thus, there is very little literature describing the impacts of working cats on rodent populations around farm buildings. However, the dairy farmers in our study were adamant that subjectively, the cats were highly effective in reducing rodent numbers, with some farmers stating they hardly see rats or mice now that they have obtained cats, and objectively, that damage and costs associated with damage to electrical wiring have been virtually eliminated. Farmers in our study overwhelmingly agreed that cats provide a more efficient alternative for rodent control on dairy farms than baits, shooting or other methods.

Reducing rodent numbers is listed as a way to decrease risk to food safety [71,101]. We recognize that domestic cats can also pose a risk to food safety through the spread of zoonotic pathogens [102,103]. Domestic cats are vectors of *Toxoplasma gondii* (*T. gondii*) and are definitive hosts of sarcocystis which can be a potential threat to livestock health [104]; however, many species of animals are vectors for pathogens, so it is difficult to eliminate the risk entirely [95,105]. Additionally, the risk cats pose to food safety can potentially be mitigated if certain measures such as sterilization and the provision of food are put in place. The sterilization of working cats reduces cat numbers over time and reduces the number of kittens and cats under one year of age [106]. This would decrease environmental contamination with toxoplasmosis oocysts, as younger cats are the primary shedders of these oocysts [107,108]. Additionally, the provision of food would reduce working cats’ need to consume rodents, which are a source of toxoplasma and sarcoystis infection for cats [109,110], without interfering with the presence of cats being a deterrent for rodents. These measures would potentially break the cycle of toxoplasma and sarcosystis infection in cats. Domestic cats are hosts for *Sarcocystis hirsuta* which is only mildly pathogenic to cattle in comparison to *Sarcocystis cruzi*, which is highly pathogenic and passed to cattle via dogs [110]. Additionally, when cats are infected with sarcocystis, they shed a low number of sporocysts [104], limiting the likelihood of infection spreading to cattle if cat numbers are controlled using measures such as sterilization.

Many participants indicated that the reduction in rats and mice around the farm led to a decrease in snakes.


*‘So it just makes my life easy because they just wander in after the morning milking [and] after the afternoon milking, they have a drink and then they disappear and I’ve got no rats. I’ve got no mice. I’ve got no snakes.’*



*‘They [the cats] do a great job. We have no mice, no rats, only a very occasional snake around the dairy.’*



*‘You see a lot of farms that have snakes hanging around, but we’ve got no snakes.’*


Although one participant indicated that they had seen the cats chasing snakes, the reduction in snakes around the dairy is more likely to be due to the decrease in the rodent population and therefore a lack of food availability for snakes. Whilst some argue that snakes should be tolerated on farms as a form of pest control [72], many of the farmers we spoke to had young children, working dogs, and domestic pets on their farm, which is likely to increase concern over having a large population of snakes. They also mentioned the risk to farmhands from venomous snakes, for example, when removing plastic covers from silage. Additionally, before having cats on the farms, farmers discussed having an overpopulation of both rodents and snakes, indicating that the snake population was not reducing the rodent population; rather, the rodent population was increasing the snake population. This is often the case with natural predators; the prey drives the abundance of the predator, not vice versa [99].

#### 3.2.2. Monetary Benefits

Several participants discussed the fact that the reduction in pests such as rodents led to a reduction in damage to infrastructure, with some indicating that they had no damage caused by rats or mice since having the cats on the farm. Participants indicated that this reduction in rodent pests had saved them money through decreased technical repairs, machinery downtime and rodenticide use.


*‘We would normally have to pay for someone to come and put those baits out because they are in the roof of the dairy. So, we’ve got the saving from a contractor and the saving of the product itself and then we’ve, I guess, the peace of mind as well that we haven’t got the potential contamination to the cows or to the milk or to dogs or to anything else.’*



*‘We haven’t had another machine eaten [by mice] since the cats have been here.’*



*‘Well, the benefits of not having any breakdowns in our electricals, because we haven’t had a breakdown in seven years since the cats turned up. So, I mean that on its own is worth thousands, plus no [downtime] with the milking machines out of action. So, yeah, the pluses are just massive.’*



*‘Well, look, it doesn’t just save you money, it saves you the hindrance of not having your machines working, because what would happen is we would get to the dairy and we’d have three or four sets of our clusters not working because they chewed some wire somewhere that was stopping those three or four clusters, you know. Yeah, so the money is one thing, but [the] pest of not being able to use your machinery is even worse.’*


Farmers previously discussed the expense of using rodenticide baits [15]. Additionally, for those using professional exterminators, the costs would likely be even higher. Therefore, no longer having to use rodenticide baits as a form of pest management is likely to have a positive economic impact on farmers. Whilst dairy farmers in Australia made record-breaking profits last year, dairy farms in QLD and NSW have the lowest rate of return of any states in Australia [111]. As our interviews were conducted with farmers in QLD and NSW, this is likely why economic saving was highlighted as a benefit to having cats on the farm. Additionally, QLD and NSW were worst impacted by the 2017–2019 drought, and NSW was the most severely impacted by the 2019–2020 Black Summer Fires [112,113]. It is likely that the loss of livestock, infrastructure and human life from drought and fires has had an impact on farming productivity, leading to farmers in the worst impacted areas placing even higher value on the monetary savings that working cats provide.

#### 3.2.3. Companionship

Whilst some participants indicated that they had little to no interaction with the cats, as they were there to do a job, others indicated that they felt a sense of companionship with the cats and enjoyed being around them. Farmers indicated that they enjoyed the company of the cats, particularly when they were milking alone in the early hours of the morning. Several participants stated that they took time out of their day to pat, watch and talk to the cats, creating positive emotions for the participants. Some participants discussed having favorites with names, with these cats coming when called.


*‘I think it’s… it’s something with cats [they] enjoy the human company. We enjoy their company as well at the dairy. Someone to have a chat to when you’re there by yourself.’*



*‘I enjoy having them [the cats] around. It’s that extra, when you get here at three o’clock in the morning, you’re on your own, like it’s… I don’t know, I talk to them, or we all talk to them. So, it’s just sort of… they’re company, I guess.’*



*‘We can watch them jump and play with each other, so I find that good to take 5 min out and be entertained.’*


Loneliness is a risk factor for farmers, leading to poor mental health, burnout and suicide [114,115,116,117]. Whilst there is little research conducted on the benefits of companionship between farmers and working animals, human–animal interactions between people and companion animals have been shown to reduce levels of loneliness [118,119]. Considering this, and the responses from farmers during interviews, we suggest that having cats around the dairy may help to combat feelings of loneliness, reducing risks posed to farmers’ mental health. Additionally, most farmers claimed ownership of the cats, although they were not microchipped and registered. Whilst mandatory registration was repealed in QLD in 2013 [120], it is a requirement for owned cats in NSW. The cost to register multiple unsterilized cats is high in NSW (A$96 annually and A$68 life-time registration fee per cat) [42,121]. Moreover, owners wishing to keep more than 2 to 4 cats are required by councils to obtain an excess permit which ranges from A$50 to more than A$500, and in some councils, annual renewal fees exceed A$300, particularly for households with more than five cats [43,44,45]. In some councils such as Ipswich, residents are also required to pay a fee of A$72 to replace a registered cat on the permit and an additional fee of A$113 to add extra cats [44]. These high costs create barriers for farmers who are already unable to pay the costs of sterilization or to microchip and register multiple unsterilized cats. 

There is a lack of clarity in the definitions of different cat populations, with the Australian Government’s 2024 Threat Abatement Plan recommending cats be classified as pet or feral [122], whereas the RSPCA’s Identifying Best Practice Domestic Cat Management in Australia report (2018) recommends classification as domestic and feral, with domestic cats defined as receiving food intentionally or unintentionally from humans and being further subdivided into owned, semi-owned (cats are fed or provided with other care by people who do not consider they own them) or unowned (receiving food unintentionally from humans) [123]. RSPCA defines feral cats as living and reproducing independently from humans [123]. There is an urgent need for feral cats to be clearly defined in legislation because this can have profound consequences for the cat’s treatment and farmers’ options for management of their cats. Feral cats are typically considered a pest species and managed by lethal means, for example, poisoning or shooting [122]. Cats around farm buildings are evidently not feral cats, nor are they pet cats, and are best aligned with RSPCA’s definition of domestic cats (owned, semi-owned and unowned). Our study demonstrates the beneficial impact working cats have on the farmers, not just in the form of pest control but also companionship, which is like working dogs or horses.

#### 3.2.4. Monetary Concerns for Care of Cats

When discussing financial considerations, participants indicated that, whilst the cats were saving them money, cost continued to be a barrier to providing care, such as sterilization for the cats. Some participants stated that they were less likely to access veterinary care as they could not visualize the return on investing in veterinary care for cats.


*‘I think as farmers, because we’re so busy and cost is a big thing for us. So, it was going to cost us a lot, and [if] it’s not putting milk into the vat, we’re not really going to do it.’*



*‘When you got 10 or 20 cats and I don’t know, it’s 250 bucks a cat or something like, I just couldn’t justify paying that sort of money to get cats desexed [sterilized], you know, so we… we just didn’t do it.’*



*‘I guess on the other side of it, if the cats aren’t classified like the dogs, they’re not working, [and] then we can’t claim them. So, I guess […] where does the money come from for that? Like I don’t know.’*


However, despite their concern about the cost, most participants indicated that they wanted to make sure the cats were cared for, with all participants at least providing the cats with milk. Most participants also fed the cats either cat or dog food and some provided veterinary care.


*‘I look after their tick thing. I have thought about not treating them for ticks because they are just working cats, but I have spoken to the local Dairy Australia vet and she said if you like them, you need to treat them because the ticks are really bad around here.’*



*‘Well, we’ll always look after them, and if we see something wrong with one of them, we don’t just go oh, well, that’s the dairy cat [so it] doesn’t get tended to. It gets looked after, yeah.’*



*‘Like if we think that they need topping up with food and stuff, we do feed them. But when the mice are high numbers, they don’t seem to come looking for food, they’re happy.’*


Whilst dairy farmers value cats due to the monetary savings associated with them, the need to reduce running costs on farms may prevent farmers from providing veterinary care, such as sterilization and vaccinations. Cost is known to be a barrier to veterinary care [124,125,126,127], and veterinary costs in Australia have been increasing at a rapid rate [128]. Additionally, food has been indicated to be the highest pet-related expenditure [129], suggesting feeding multiple cats on farms may be a large financial burden to farmers. This, alongside the small profit margins for dairy farmers in QLD and NSW, as well as the economic impacts of recent natural disasters, is likely to reduce the prioritization of veterinary care and food for cats on farms. However, some farmers provided veterinary care despite concerns about cost. People have been shown to be more willing to buy products if they have high functional and emotional value [130]. This could be applied to the value dairy farmers place on working cats, specifically that farmers who did provide care consider the high functional and emotional value of the cats and the service they provide. Public concerns have been raised regarding the welfare of free-roaming cats. However, when cats are provided with care measures such as sterilization, vaccinations and food, this can improve and maintain their welfare [131,132]. Therefore, making veterinary care more accessible and educating farmers on the positive implications of sterilizing, vaccinating and feeding cats may help to promote the provision of care, improving cats’ welfare whilst reducing their environmental impacts and financial strain on farmers.

### 3.3. Classifying Cats as Working Animals

The outcomes of the thematic analysis indicated that the farmers strongly supported the ATO classifying cats on farms as working animals, and that they were able to generate a number of arguments as to why cats should be classified as working animals. These are shown below and have been tabulated with context examples in Table 3.

#### 3.3.1. Farmers’ Positive Views

All participants were supportive of cats on farms being classified as working animals. One participant had a positive view but suggested that there should be a limit on the number of cats that can be registered as a tax deduction. Others discussed the idea that making the cats’ care tax deductible may improve the cats’ welfare, as it would be an incentive to provide veterinary care and would help to relieve the financial burden and stress of caring for the cats. Additionally, many participants stated that classifying the cats as working animals was a good idea as the cats were ‘doing a job’.


*‘I think it’s excellent, as long as it’s not […] someone that’s got 200 [cats], possibly a limit needs to be put on it.’*



*‘I think it would also improve the welfare of the animals because I think whether they are tax deductible or not, we still would have them, but knowing that anything that we spent on those animals, if that was tax deductible, then that’s another incentive to do it.’*



*‘That would make it a real advantage for a farmer. [A] massive advantage, because we’ve already got enough costs, and we don’t need any more.’*


The literature supports farmers’ comments indicating that the provision of care, such as sterilization, vaccinations, food, and flea, tick and worm treatment, improves cat welfare [15,133,134,135]. Additionally, sterilization leads to a reduction in population size and has been suggested to reduce hunting behaviors and roaming distances [15,133]. Moreover, sterilization would reduce the number of kittens and lactating mothers. This is likely to reduce cat predation on wildlife, as young cats show more hunting behaviors [106,136], and lactating cats have higher energy requirements, increasing the amount they must consume to meet these requirements [137] and therefore increasing their need to hunt. The sterilization of cats would also reduce risks to food safety by stopping the increase and potential overpopulation of cats and by reducing the number of kittens and cats under one year of age. This would decrease environmental contamination with toxoplasmosis oocysts, as younger cats are the primary shedders of these oocysts [107,108]. A need for food is likely to drive the migration of cats. Therefore, providing cats with food would also reduce the risk of cats relocating to bushland when the rodent population diminishes, mitigating their impact on wildlife populations. Providing sufficient food to meet the cats’ nutritional requirements would also reduce their need to consume rodents, without interfering with the presence of cats being a deterrent for rodents, which could help to break the cycle of toxoplasmosis infection in the cats. Provision of care for working cats would improve their welfare and reduce their impact on wildlife and food safety. With financial pressures already a stress factor for farmers [28,60,63,116], permitting working cats’ care to be tax deductible will not only act as an incentive to provide care, thus improving cat welfare and reducing their impact on the environment, but would be expected to reduce associated risk factors for farmers’ mental health.

#### 3.3.2. Farmers’ Arguments to the ATO About the Importance of Cats on Farms

Participants were asked how they would argue to the tax office that the cats should be classified as working animals. There were a variety of arguments made, with some stating that rodenticides were tax deductible, so the cats should also be tax deductible as they are serving the same purpose. Others compared the cats to animals already classified as working animals, indicating that the cats were doing an equally important job as working dogs and horses, with one participant stating that the cats were doing more work than his working dog.


*‘You probably have to compare yourself to the farmers [who] use other methods for control like poisons and that sort of thing. Like poisons [for] vermin control are a tax deduction, so why can’t the cats be a tax deduction as well, because they’re doing the same things as the poisons?’*



*‘But [the cats are] definitely doing a job and a purpose and it’s probably more important than some working dogs.’*



*‘Well, when we do our food safe audit, […] if you put baits out, you have to tell them when you put baits out like […] blah blah, but we… we literally write, we have cats, they’re our vermin control. That’s all we have to state and that classes for our food safe handling. So, what’s the difference really?’*


In other countries, such as the UK and USA, these arguments would be enough to allow cats on farms to be a tax-deductible business expense [38,39,41]. However, the ATO only permits dogs and horses to be classified as working animals and has specific criteria to be met, including that ‘the animals must be trained for their role from a young age and must not be treated as pets’ [37]. Whilst our study has shown clearly that cats on dairy farms are viewed primarily as working animals and not as pets, the criterion for training is problematic because cats being used for pest control do not need to be trained as they are natural predators of rodents. The ATO needs to redefine their criteria for working animals and recognize cats’ roles in pest management on farms. Permitting their care to be tax deductible would enable farmers to practice responsible ownership without putting further financial strain on them. This would allow farmers to have the benefit of cats as pest control whilst reducing their impact on the environment, including wildlife, and improving food safety. It is also recommended that state government legislation be amended so that cat registration and permit costs be waived, as they are for working dogs on farms.

### 3.4. Limitations

Only farmers who had working cats and participated in the free cat sterilization program were interviewed. We recognize the small sample size of this study (15 interviews across nine farms in two regions), which represents 0.2% of dairy farms [138] and 0.01% of all farms [139]. However, we targeted farmers who had participated in a free working cat sterilization program and interviewed farmers from 90% of farms (9/10) that had participated in the pilot program. Therefore, our small sample size is an appropriate representation of this small population. We were interested in exploring an issue that remains poorly understood and documented in Australia and in providing baseline insights to inform more detailed future studies. Previous qualitative research of a similar nature used a small sample size to gain in-depth and rich understanding of a topic [15,140]. However, the number of farmers who participated in a free working cat sterilization program is a small proportion of the farming population of Australia. We recommend that the results of our study are used as a basis for future research to build upon, including identifying other types of primary producers who maintain a population of cats for rodent control.

## 4. Conclusions

This study confirms the well-documented negative impacts of rodent overpopulation. We hypothesize that the negative impacts of rodent overpopulation may be more substantial on dairy farms than elsewhere, because of the automated, complex nature of the machinery used for milking and the resultant welfare issues if cows cannot be milked due to equipment failure. However, further research should be conducted to evaluate the impact of rodents on various types of farms, and to determine whether there is a relationship between the complexity of farm machinery and the economic cost of rodent-related damage. Nevertheless, the views shared with us demonstrate that the dairy farmers in the study valued the use of working cats on their farms to suppress rodent populations and prevent associated equipment damage and loss of produce. Our research indicates that farmers place a high value on cats due to their efficiency in pest control and associated monetary savings. However, because of cost, farmers face barriers to providing care measures such as sterilization and vaccinations, as well as abiding by state and local government laws relating to microchipping, registering and obtaining relevant permits associated with the cats. We recognize that an uncontrolled population of free-roaming cats can have just as detrimental an impact on wildlife populations and food safety as an overpopulation of rodents. We recommend that federal and state government funding, in collaboration with support from industry bodies and animal welfare organizations, be provided to assist farmers in reducing cat numbers around farm buildings through sterilization, and that the ATO permit cats on farms to be classified as working animals, allowing their care to be tax deductible and to facilitate ongoing population control through sterilization. Dairy farmers indicated that this would enable them to sterilize cats to manage the population. Such measures would likely contribute to a decrease in the cat population and improve cat welfare and food safety, whilst helping to protect the environment. State and local government registration and permit costs for working cats should also be waived, as they are for working dogs on farms. Until we recognize the value of cats in people’s lives and work with them to manage their cats in an assistive way, we will not be successful in reducing the cat overpopulation and the associated negative impacts.

## Figures and Tables

**Table 1 animals-15-00800-t001:** Pest control issues, themes and sub-themes with context examples from interview transcripts.

Themes	Sub-Themes	Context Examples
Damage to Infrastructure and Produce by Rodents	Machinery breakdowns Damage to stock feed	‘Well, basically the wiring had to be replaced all the time. Every two months he was coming to replace wiring that had been eaten.’ ‘I would say at least two or three thousand dollars a year, trying to fix those things [wiring chewed by rats]. It could be a lot more, but as an average year, that would be easy.’ ‘And in the hay shed as well, like the mice, if they take over the cereal hay crop in the hay shed, [they] just demolish it.’
Human and Animal Health Issues	Human health and safety Livestock health and safety	‘I guess it’s a human health thing too, like with the milk and everything that’s around, we don’t want rodents.’ ‘Another place that we had them [rodents] in droves was at our silage pit facility, which was encouraging snakes.’ ‘With staff, you don’t want snakes around.’
Dislike of Rodenticides	Inefficient Expensive Concern about impact on wildlife Safety of pets and children	‘Well, like I said, we do have some baits in certain sheds, but we just find, yeah, sometimes the rats prefer to chew what they’re not supposed to chew [rather] than the bait.’ ‘If they [the rats] do go and die in the cows’ feed, that is a huge health concern for the cows [if] we’re putting up feeds that have had decomposing mice that had been poisoned.’ ‘Yeah, we were using, yeah, rat bait most of the time and that was costly as well. […] Yeah, you know, we spent a couple hundred… two or three hundred dollars year, no trouble.’

**Table 2 animals-15-00800-t002:** The value of cats to farmers, themes and sub-themes with context examples from interview transcripts.

Themes	Sub-Themes	Context Examples
Pest Control	Efficient rodent control Preferred method of pest control Reduction in rodents and snakes	‘Well, [we have the cats] really just for the pest control, really, if it wasn’t for that, then we probably wouldn’t have any.’ ‘Yeah, we haven’t done it [used rodent baits] for probably 4 years now, yeah since we’ve had cats.’ ‘Well, we just baited them [the rodents], but they were still there, the baits weren’t doing the job, not like cats.’ ‘Since the mice and the rat population disappeared, I reckon we’ve got a lot less snakes hanging around the sheds.’
Monetary Benefits	Cats reduced machinery repair and downtime costs Cats reduced costs associated with rodenticides	‘Less electrician costs, yeah, less damage to other [equipment] that they [rats and mice] would chew and eat or nest in.’ ‘I would say they [the cats] were saving us between two and three thousand dollars a year. […] At least.’ ‘We need some cats at the dairy because we just keep having problems with our wiring, cause the rats keep getting in and chewing them. Yeah, cats are cheaper than an electrician bill.’
Companionship	Positive interactions Positive emotions	‘They’re sort of like a mate, so you… you sort of think, yeah, when they come in, they meow to you, and you go out and you feed them and give them a pat.’ ‘Yeah, it gives you a bit of a bit of a buzz, I guess with them being all waiting around and waiting for their little drink of the morning and yeah, so that… that’s a good thing.’
Monetary Concerns for Care of Cats	Cost as a barrier to providing care Some care provided	‘The cost is too great to have to get all those cats done [sterilized] ourselves.’ ‘There is always like dry dog food biscuits open [for the cats] that ad-lib whoever wants them. I have seen them in there [eating dog food], but other than that, they just get their milk, yeah, that’s all we provide for them.’ ‘They get flea and wormed but probably only once a year, whenever we get around to doing it, if that makes sense.’

**Table 3 animals-15-00800-t003:** Classifying cats as working animals, themes and sub-themes with context examples from interview transcripts.

Themes	Sub-Themes	Context Examples
Farmers’ Positive Views	Reduce financial strain on farmers Improve cat welfare	‘Yeah, and I think [the] majority of farms, if it was a deductible expense, they’d probably buy a bit more tick treatment because they go, oh yeah, it’s a business expense.’ ‘They [farmers] need cats around that area, around the hay shed to do what they need to do around the barn. Yeah, so I think if you’re classed as a prime producer, you should have the full right to be able to claim the cat deductions.’
Farmers’ Argument to the ATO about the Importance of Cats on Farms	Providing same service as rodenticides Just as important as other working animals	‘I mean [the cats are] doing the same job as a dog on the farm or a horse on the farm, they’ve got a role. […] We need them on the farm, so yeah, they’re a tool that we use on the farm.’ ‘They’re dead set working cats because of the pointers that I’ve pointed out; the saving on repairs, the saving on baiting and yeah, the cats are doing their job, they’re basically working for me.’

## Data Availability

Most relevant data are reproduced in the text.

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
