# Peer review of "Feline Farmhands: The Value of Working Cats to Australian Dairy Farmers—A Case for Tax Deductibility"

_animals, 2025, doi:10.3390/ani15060800_

Round 1
Reviewer 1 Report
Comments and Suggestions for Authors
I found this this manuscript to be an interesting exploration of Australian dairy farmers’ views of working cats and the need for working cat care to be tax deductible. Utilizing semi-structured interviews, the authors attempted to gain a deeper understanding of fifteen dairy farmers’ perspectives on the benefit and cost of working cats. The dairy farmers, from nine different farms, were recruited from two free sterilization programs and asked a series of questions about their use of working cats. The authors used thematic analysis on the transcribed interviews. The analysis resulted in nine main themes. The authors found that the farmers felt that rodents caused infrastructure damage and health concerns, and they disliked rodenticides. Additionally, the farmers mentioned that the cats were effective at pest control, they provided companionship, and although the cats saved the farmers money, they also had concerns about funding cat care. The farmers also felt that making working cats tax-deductible would reduce financial burden and improve cat welfare. Overall, the study suggests that Australian dairy farmers believe that barn cats are working animals because they provide a critical service to the dairy industry, and their care should be considered tax-exempt like other working animals.
I enjoyed your manuscript! I think it makes a strong argument for Australia to consider cats working on dairy farms as working animals and thus, tax-exempt.
Simple Summary
|
Line 18 |
I suggest changing “dogs” to “working dogs.” It currently reads as a little redundant because you have pets later in the sentence. |
Introduction
|
Lines 60-61 |
You mention “rodents’ gnawing habits can cause damage to wiring and other expensive infrastructure,” but there’s no citation. Is there any data to support this? |
|
Lines 64-72 |
It might be worth mentioning that these methods of rodent control are tax deductible. One (or more) of your farmers mention it in their statements and that seems relevant to your argument. |
|
Lines 110-120 |
This is quite redundant. I suggest removing it. |
Methods
|
Lines 155-156 |
Who estimated their age? The interviewer? I’m not sure this is appropriate to include since it’s the interviewer’s opinion. Is this how gender was determined? Or were farmers asked how they identify? If they were not given the chance to self-identify, then I think you should remove that section too. |
|
Lines 157-160 |
Does the mean number of cats reflect the mean across farms? Or the mean across participants? You had multiple participants from the same farm, so this could skew the mean. |
|
Lines 165-166 |
Did participants have to agree to the interview to receive the free sterilizations? I would explicitly mention it either way. |
|
Lines 181-187 |
Might be worth mentioning why an author’s phone number was used for the cats’ microchips. |
|
Line 207 |
Did the duration of the interview include questions that were not used in this study? |
|
Lines 211-215 |
How many questions were there? I think it would be useful to explicitly state that. |
|
Lines 216-227 |
There is not nearly enough detail on how the thematic analysis was conducted. Was each question analyzed on its own? You mention nine main themes, but it seems like certain themes only pertained to certain questions. How long was each unit of meaning? Was there any double coding? How was the validity of the coding determined? Please expand this section. |
Results and Discussion
|
Table 1 (and future tables) |
It would be useful to include who said each quote. I suggest including something like “Participant 3, Farm 2.”
|
|
Table 1 |
How are machinery breakdowns and damage to equipment separate sub-themes? It is unclear how they differ. I suggest only having two sub-themes for Theme 1 or providing context examples that justify why they need to be different. |
|
Lines 239-240 |
How many farmers had previously been without cats? That seems like a relevant information to share. They likely have not had as much damage from rodents and have not used rodenticides before. Thus, they might not be represented in some of the questions. |
|
Table 1 |
It is confusing that food safety is its own sub-theme as food safety is an aspect of both animal and human health. |
|
First box under damage to infrastructure |
Is the third quote related to infrastructure? It also seems relevant to animal health and safety/ food safety. Perhaps it shouldn’t be used as an exemplary example. |
|
Lines 258-264 and all over |
Cost is mentioned so many times throughout the paper and quotes you provided. It’s surprising that cost did not come out as a theme. |
|
Table 2 |
How is “Cats provide” a sub-theme? Cats provide what? |
|
Table 2 |
The theme (monetary benefits) and sub-theme (economic savings) are the same thing. |
|
Lines 461-469 |
I really like your comments about the need for better definitions around feral cats and the different types of cats in general. I think this is very important. |
|
Table 3 |
Are “Farmers’ Views” and “Farmers’ Arguments” really themes? They’re not really overarching ideas, but more like organizational units. |
|
Lines 574-577 |
It might be worth mentioning that a potential limitation is that you only interviewed dairy farmers who 1) had cats, and 2) wanted access to free sterilization. This might not be representative of all dairy farmers in Australia. |
Reviewer 2 Report
Comments and Suggestions for Authors
I went with a strong interest trough this proposal of making cats' assistance and care tax deductible for AUS dairy farmers. To date, the most significant achievements in the field of animal welfare have been mainly the results of the voluntary improvements adopted by animal owners and keepers - driven by citizens and consumers - and the rigorous enforcement of legislative requirements, being them both expected as structural elements of responsible animal ownership.
However, I have some concerns, besides the very the low number of dairy farms involved (only nine out of a total number of 744 farms registered in 2023, with 278 in QLD and 466 in NSW) in the study, already identified as a limitation of the study.
In particular, it would be important if the Authors could clarify and elaborate on:
- The Australian legislative approach to the managements of cat's welfare, at national, state and local level;
- The eligibility requirements for the farmers enrolled in the sterilization programs;
- The level of implementation of the Animal Care and Protection Act 2001 in these dairy farms, considering that specific norms are already in force in QLD, establishing that a person in charge of an animal owes a duty of care and should take reasonable steps to provide it with food and water, accommodation and living conditions allowing to display a normal behavior. In this frame, animal owners or keepers should guarantee the treatment of disease or injuries in “peace time” and during natural disasters. Similar rules are in place in NSW, where companion animals’ welfare is protected under the Prevention of Cruelty to Animals Act 1979 (POCTA), also establishing that good animal welfare requires disease prevention and veterinary treatment, appropriate shelter, management, nutrition and humane handling. People involved in the care and welfare of cats must comply with these norms.
Reviewer 3 Report
Comments and Suggestions for Authors
This manuscript presents the “thoughts and feelings” of 15 dairy farmers on the possible impacts of roaming cats on rodent control for their farm. It also presents their views on cats being classified as “working cats” from a tax perspective, thereby providing financial benefits to those farmers interviewed. The results are presented as summaries interspersed with anonymous quotes from the interviews and then interpretation of these findings.
I have significant concerns with the way the results of this manuscript are framed and interpreted. The conclusions drawn from this manuscript are at odds with the literature, ignore (or underplay) significant issues of relevance while at the same time raising obscure issues of minor importance, and over-generalise the findings. The following four issues are the primary factors that I will focus on:
· Most importantly, the manuscript never questions the primary premise presented – that cats are an effective control mechanism for rodents. The reality is, however, that the ability of free-ranging cats to effectively control pest rodent populations is unsubstantiated. I have included at the end of this review a number of recent publications (including reviews) that support this finding and there are many more examples that could be cited. In a way, this reality sets up the manuscript to be a house of cards from the beginning, particularly because there is no balanced critique to appropriately contextualise the assertions from the farmers. As such, their misleading claims are left unchallenged and are even taken as fact. An opportunity to prevent misinformation is passed up and instead conclusions are presented to act on these often inappropriate assertions.
· The manuscript presents a limited, relatively one-dimensional framing of alternative mechanisms for controlling rodents on farms, particularly rodenticides. While I agree that there needs to be increased caution around the use of second generation rodent baits, the very negative framing of bait effectiveness needs to be more balanced and better reflect the literature. While this many not help build the case for using cats as rodent control, it will help to present a less biased manuscript.
· I understand that the design used (a phenomenological approach) can gather useful insight to understand the views of participants. However, when this data is not appropriately interpreted with reference to current literature and significant issues relevant to the matters raised, it can lead to significant bias and inappropriate conclusions. It is my view that this lack of balanced interpretation is a major issue with this manuscript. In addition to the lack of a balanced framing of cats being an effective control mechanism for rodents (see point above), the manuscript either presents a very limited framing or overlooks discussing the findings in light of the broad literature on (1) the negative welfare outcomes for free-roaming cats, (2) impacts of free-roaming cats on public and environmental health, and (3) impacts of free-roaming cats on livestock health. I note, for example, that the human and animal health issues mentioned (from line 284) focus only on the threat rodents represent. Please note that I am not referring to the absence of these issues in the farmer’s feedback – their opinion should be theirs alone – but rather in the interpretation of their feedback as it is summarised for the reader.
· The ‘Conclusions’ section is not consistent with the evidence presented, being a series of non sequiturs. It is misrepresentative of both the findings of the manuscript and the wider literature as follows:
o The manuscript presents no findings at all that show dairy farms have greater impacts from rodents “than elsewhere” (line 586), nor any rationale that it is because of milking machinery or operations (over other farming machinery or operations). The conclusion “the value of working-cats to dairy farmers cannot be underestimated” (line 589) is completely unsupported.
o “Our research indicates that farmers place a high value on cats, due to their efficiency in pest control and associated monetary savings” is again unsupported by the findings presented and the wider literature – these are merely unsupported assertions (without evidence) by a very small number of farmers.
o “We recognize that an uncontrolled population of free-roaming cats can have just as detrimental an impact on wildlife populations and food safety as an overpopulation of rodents. We therefore recommend that the ATO remove barriers to farmers by permitting cats on farms to be classified as working animals, allowing their care to be tax deductible.” Again, this is a non sequitur – just because free-roaming cats can impact wildlife populations and food safety does not at all justify that cat care should be tax deductable for farmers.
o “Until we recognize the value of cats in people’s lives and work with them to manage the cats in an assistive way, we will not be successful in reducing numbers and protecting wildlife.” This assertion is not at all backed by evidence, particularly the assertion regarding protecting wildlife.
Finally, I note that there are multiple other lesser issues that will need to be worked on if the manuscript is to be resubmitted for publication. I will not provide feedback on each individual section, given the likely changes that will be required, but four issues are covered here:
· I have some concerns about the study design. How is it appropriate to consider participants from the same farm as independent contributors (there were only 9 farms and there were 15 interviews)? How can people in the same interview (two interviews had two people in them) be considered as independent contributors, assuming they could hear each other’s answers?
· How is it reasonable to assert that 60 cats on a single farm (line 159) are all there for the purpose of “pest control” (line 349)? Surely this is an opportunity to provide further context for the reader as to the dubious validity regarding the assumptions of the farmers on what their cats actually do on their properties?
· The manuscript states that a small sample size is possibly limiting (line 570, “We recognize the small sample size of this study”), yet completely overstate the findings in the summary by stating (line 22) “The results demonstrate that working-cats play a vital role in pest control on dairy farms…”
· The manuscript states (line 394) “Additionally, the risk cats pose to food safety can potentially be mitigated if certain measures are put in place.” No references are supported to back this dubious assertion and the manuscript completely ignores expanding on this critical issue.
Example references countering the claims that free roaming cats can be an effective rodent control:
Koizumi, R., Endo, T., Tanikawa, T., Hirata, S., & Kiyokawa, Y. (2024). Coexistence of roof rats and carnivores in barns on a livestock farm in Japan. Animal Science Journal, 95(1), e13982. https://doi.org/10.1111/asj.13982
Krijger, I. M., Gort, G., Belmain, S. R., Groot Koerkamp, P. W. G., Shafali, R. B., & Meerburg, B. G. (2020). Efficacy of Management and Monitoring Methods to Prevent Post-Harvest Losses Caused by Rodents. Animals, 10(9), 1612. https://doi.org/10.3390/ani10091612
Calver, M. C., Cherkassky, L., Cove, M. V., Fleming, P. A., Lepczyk, C. A., Longcore, T., Marzluff, J., Rich, C., & Sizemore, G. (2023). The animal welfare, environmental impact, pest control functions, and disease effects of free-ranging cats can be generalized and all are grounds for humanely reducing their numbers. Conservation Science and Practice, 5(10), e13018. https://doi.org/10.1111/csp2.13018
Parsons MH, Banks PB, Deutsch MA and Munshi-South J (2018) Temporal and Space-Use Changes by Rats in Response to Predation by Feral Cats in an Urban Ecosystem. Front. Ecol. Evol. 6:146. doi: 10.3389/fevo.2018.00146
Round 2
Reviewer 1 Report
Comments and Suggestions for Authors
I thank the authors for clarifying the methods they used for thematic analysis and for addressing my other suggestions. I think the manuscript looks good and I look forward to seeing it published. Great work!
Reviewer 2 Report
Comments and Suggestions for Authors
Thanks for addressing the critical points highlighted in my previous review.
As I understand, feral cats are exempted by animal protection norms and the control of cats have little animal welfare protection for any cats deemed feral. However, "most farmers claimed ownership of the cats, although they were not microchipped and registered", therefore these animal should be not recruited for the study.
In addition, you should also express your opinion in favour of the farmers who abide by the rules with the justification that they not have the resources to comply with legislation or enforcement orders. Actually,it has been documented that the mental attitude of the owners is one of the biggest obstacles to the spread of sexual neutering of companion animals, and not the lack of financial resources (Fournier A, Geller S. Behavior Analysis of Companion-Animal Overpopulation: A Conceptualization of the Problem and Suggestions for Intervention. Behaviour and Social Issues. 13, 51–69, 2004. doi.org/10.5210/bsi.v13i1.35.).
If you really believe that an assistive rather than an enforcement way will better guarantee the sustainability of cat population management in Australia, you should provide evidences on the efficacy and efficiency of this approach taken somewhere, which I personally see pretty in contrast with the Responsible Ownership concept, even more in this context where cats are contributing to the farming economy.
Reviewer 3 Report
Comments and Suggestions for Authors
This review covers the revised manuscript dealing with the thoughts and feelings of a small group of dairy farmers regarding the possible impact of roaming cats on rodent control for their farm. I have considered both the revised manuscript and the response to my earlier feedback in this review.
My earlier concerns with the way that the results of this manuscript are framed and interpreted have not been addressed adequately. In fact, it is my view that changes made in response to the first round of feedback have made the manuscript more polarised and biased in its interpretation and presentation of the relevant literature. In the last review I included a small selection of literature that provides a way into the far broader context required to present unbiased framing to the topics being discussed. I am disappointed that the revised manuscript did not take the opportunity to reconsider how the study was framed (e.g. only two of the four papers provided are cited, and both in a cursory off-hand way, with no additional papers included to balance the framing).
In fact, it seems that this point, what I considered the most important point of the review, was entirely missing from the response to reviewer letter, so I will include it again:
“Most importantly, the manuscript never questions the primary premise presented – that cats are an effective control mechanism for rodents. The reality is, however, that the ability of free-ranging cats to effectively control pest rodent populations is unsubstantiated. I have included at the end of this review a number of recent publications (including reviews) that support this finding and there are many more examples that could be cited. In a way, this reality sets up the manuscript to be a house of cards from the beginning, particularly because there is no balanced critique to appropriately contextualise the assertions from the farmers. As such, their misleading claims are left unchallenged and are even taken as fact. An opportunity to prevent misinformation is passed up and instead conclusions are presented to act on these often inappropriate assertions.”
I return to my point above raised in the first review: there is very little balanced critique to appropriately contextualise the assertions from the farmers. As such, their misleading claims with next to no supporting evidence presented are left unchallenged and are even taken as fact. An opportunity to prevent misinformation is passed up and instead conclusions are presented to act on these often-inappropriate assertions. It is this framing that concerns me most about this manuscript and why I still have real concerns with its publication in its current form.
At this point I could go through and summarise section by section how I still think that the earlier material remains unbalanced, that the added material (with the exception of the additional content regarding livestock health, which I thought was very well done) has compounded that bias, and suggest ways to improve the document, including references. However, I have no evidence from how the most recent revision was handled that this will produce change, so I’m reluctant to spend time generating such content.
That said, I think it is important to get research published and I would like to find a solution here, even if I do not support publishing this manuscript in its current form. I like the fact that this manuscript focuses on an area where we have little insight, at least in Australia. I also recognise that the topic of this manuscript is highly controversial and that there are contrasting viewpoints held by those involved with such work that are never likely to be reconciled anytime soon. We should not be looking to supress this ongoing discussion in the review process, but neither should we be enabling the publication of manuscripts that, when published, can be used to give false credibility when pushing agendas.
Given the above, it is my opinion that the most important issue to address here is to ensure that the manuscript does not over-generalise the findings and is far clearer throughout as to the severe limitations of what is only a very small pilot study at best – 9 farms and 15 interviews. DAFF data from 2024 suggests that there are over 4,500 dairy farms in Australia. Therefore, this insight represents a mere 0.2% of dairy farms from just two regions. Further, given that there are more than 87,000 farms in Australia, it is even more inappropriate to generalise these findings to all farmers, let alone make any recommendations for national tax changes!
Taken together, this context reinforces that any statements of generalisation beyond the specific study (even to the two study regions) are highly inappropriate. Based on this approach, I suggest the following changes are a bare minimum for publication:
Title: It is a big assumption that roaming cats on a farm are doing “work” and no evidence is presented in this manuscript to support that assertion. Therefore the term “working-cats” seems inappropriate, and I would suggest changing this to “roaming [or stray] cats on farms” (and reconsider the use of the term working-cats elsewhere in the manuscript). Further, in a response to my first review, it was stated “we have not stated all cats currently on farm are for pest control.” That statement reinforces the idea that “working-cats” is not an appropriate term to use. I also suggest removing the “A case for tax-deductibility” from the end as such a recommendation is well beyond the findings of this manuscript (see detailed comments below on this matter).
L20-22: Change to “This study sought the opinions of dairy farmers on the value of roaming cats on their farms and making the costs of their care tax deductable with the Australian Tax Office.”
L22: Change to: “From 15 interviews (9 farms) we conclude that farmers think that roaming cats play an important role in pest control…”
L24: Change to: “…on the farm than use poison baits, as they viewed cats as a safer, cheaper,…”
L27: It is completely beyond the findings of this manuscript to generalise the findings to support a recommendation of making the care of roaming cats on farms (particularly all farms!) tax deductable. Change to: “It may be that there is a role for roaming cats in providing some rodent control on dairy farms. Our exploratory survey found that farmers wanted the costs of their care considered as a tax deductible working expense on the grounds that there may be benefits to the well-being of farmers and cats whilst…”
L33: Change to: “…rodenticide baits, which when used incorrectly can be expensive, inefficient, and can pose a risk to children…” As it stands now, this is a significant over-simplification of the bait situation, and combines both first- and second-generation baits into the one statement, which is really not appropriate.
L35: Change to: “…but concern for cats’ impact on wildlife and negative outcomes for their own welfare from roaming may lead to them being…”
L36: Change to: “…interviews with 15 people from 9 dairy farms in two regions, we explored the value cats…”
L37: Change to: “The opinions collated claimed that uncontrolled rodent populations….”
L42-44: Again, this generalisation is completely beyond the findings of this paper. That is, the opinion of 15 people from 9 farms, representing 0.2% of dairy farms (and 0.01% of all farms) should never be used to support such a broad generalisation. Change to: “The views collated in this study suggest that there may be value in having roaming cats on dairy farms and that quality and welfare outcomes could be improved by greater resourcing. A broader evidence-based investigation of the role of roaming cats in supressing rodents on dairy farms would first be required before recommendations of funding support could be made.”
L107: I think it is overstating the case to suggest that cats are an “effective rodent management strategy”, and there is literature to support that. Change to drop the term “effective”.
L108: The claim “It may also minimise additional damage to Australia’s fragile ecosystems” is entirely unsupported and unexplained. It is also unclear as to what “it” is. Please delete this sentence entirely.
L121-126: As noted elsewhere, this claim is completely beyond the findings presented in this manuscript. I suggest deleting all of the lines completely, or otherwise replacing with “We anticipate that our findings will help to develop more evidence based studies that could, in turn, inform local government and welfare agencies how best to manage roaming cats on dairy farms specifically considering their possible role in rodent management.”
L616-623: This content is particularly under-developed and misleading. To claim that “the aim was to gain an in-depth understanding of the value of cats on farms to farmers” does not at all fit with the scope or extent of the study as mentioned previously. Further, the content of the interviews presented does not fit with an “in-depth understanding” or “rich and detailed data” – it is more along the lines of a preliminary assessment. Importantly, this is an area where there is not great understanding, at least in Australia, and I suggest instead focusing on that as a justification for the limited replication and depth. Change to: “…sterilization were interviewed. We recognise the small sample size of this study (15 interviews across 9 farms in two regions), representing 0.2% of dairy farms and 0.01% of all farms. However, we were interested to explore an issue that remains poorly understood and documented in Australia, and to provide baseline insight to inform more detailed future studies. Additionally, we targeted farmers who had participated in…”
L633-636: Given the wide variety of quotes presented in the paper from the farmers interviewed, I do not see how this one issue can be extracted as the explanatory reason why rodent impacts are greater on dairy farms. Note that I commented specifically on this issue in the first round of review. Rather than presenting it as a finding, I suggest it is flipped around and presented as a hypothesis to explore in future work. The manuscript can then be used as a way to justify explicit evidence-based testing of this hypothesis. To be clear, I don’t disagree with the hypothesis (it’s a good one), I just do not see any evidence in the paper to justify it as a conclusion.
L637-638: Change to: “Nevertheless, the views shared with us indicate that there may be value in maintaining roaming cats on dairy farms to suppress rodent populations. Our research indicates that farmers…”
L643-647: Remove entirely from “We therefore recommend…” to “…and protecting the environment.” As noted in many comments above, it is entirely beyond the scope of this manuscript to be calling for the ATO to remove barriers and providing tax deductibility for the care of cats based on the opinions (with no evidence to back these opinions) of 15 people from 9 farms.
L649: “reducing numbers” of what? Further, what does “protecting wildlife” have to do with the positive outcomes presented in this manuscript? As I see it, the only outcomes for wildlife presented are death (or habitat disruption) for snakes and predation of other wildlife from the roaming cats. Best to keep the positive framing on issues actually relevant to the scope of the manuscript.
